# Characterization of Thermal Bio-Insulation Materials Based on Oil Palm Wood: The Effect of Hybridization and Particle Size

**DOI:** 10.3390/polym13193287

**Published:** 2021-09-26

**Authors:** Indra Mawardi, Sri Aprilia, Muhammad Faisal, Samsul Rizal

**Affiliations:** 1Doctoral Program, School of Engineering, Post Graduate Program, Universitas Syiah Kuala, Banda Aceh 23111, Indonesia; indratm@pnl.ac.id; 2Department of Mechanical Engineering, Politeknik Negeri Lhokseumawe, Lhokseumawe 24301, Indonesia; 3Department of Chemical Engineering, Universitas Syiah Kuala, Banda Aceh 23111, Indonesia; sriaprilia@unsyiah.ac.id (S.A.); mfaisal@unsyiah.ac.id (M.F.); 4Department of Mechanical Engineering, Universitas Syiah Kuala, Banda Aceh 23111, Indonesia

**Keywords:** bio-insulation, hybrid panel, oil palm wood, bio-based adhesive, tapioca starch, thermal conductivity, physical and mechanical properties

## Abstract

Oil palm wood is the primary biomass waste produced from plantations, comprising up to 70% of the volume of trunks. It has been used in non-structural materials, such as plywood, lumber, and particleboard. However, one aspect has not been disclosed, namely, its use in thermal insulation materials. In this study, we investigated the thermal conductivity and the mechanical and physical properties of bio-insulation materials based on oil palm wood. The effects of hybridization and particle size on the properties of the panels were also evaluated. Oil palm wood and ramie were applied as reinforcements, and tapioca starch was applied as a bio-binder. Panels were prepared using a hot press at a temperature of 150 °C and constant pressure of 9.8 MPa. Thermal conductivity, bending strength, water absorption, dimensional stability, and thermogravimetric tests were performed to evaluate the properties of the panels. The results show that hybridization and particle size significantly affected the properties of the panels. The density and thermal conductivity of the panels were in the ranges of 0.66–0.79 g/cm^3^ and 0.067–0.154 W/mK, respectively. The least thermal conductivity, i.e., 0.067 W/mK, was obtained for the hybrid panels with coarse particles at density 0.66 g/cm^3^. The lowest water absorption (54.75%) and thickness swelling (18.18%) were found in the hybrid panels with fine particles. The observed mechanical properties were a bending strength of 11.49–18.15 MPa and a modulus of elasticity of 1864–3093 MPa. Thermogravimetric analysis showed that hybrid panels had better thermal stability than pure panels. Overall, the hybrid panels manufactured with a coarse particle size exhibited better thermal resistance and mechanical properties than did other panels. Our results show that oil palm wood wastes are a promising candidate for thermal insulation materials.

## 1. Introduction

Oil palm (*Elaeis guineensis*) is well known as one of the most significant agricultural plantation types in Indonesia, and it has a critical role in the Indonesian economy. Indonesia is one of the largest palm-oil-producing countries globally, with 48.42 million tons of crude palm oil produced from a plantation area of 14.60 Ha [1]. Each hectare can be planted with 135–145 trunks, of which each trunk generates around 10% oil palm, and the remaining 90% is solid waste [2]. The palm oil industry produces various types of waste during milling processes and plantation activities. The primary wastes generated during the milling process are palm kernel shells, mesocarp fibers, and empty fruit bunches, while the main wastes from plantations during logging are fronds (around 20–27%), trunks (70%), leaves (6.5%), and others (3%) [3]. Rejuvenation produces around 74.48 tons of trunks per hectare; with an area of 14.6 hectares, around 44 million tons of trunks will be produced [4]. Usually, a large number of these felled trunks are left to rot in the field. The volume of trunk waste is expected to increase rapidly with increasing plantation rates every year, posing a serious problem in the future. Therefore, attention and effective waste management are required to convert this waste into a value-added product.

The abundant amount of waste oil palm trunks has the potential for use in non-structural wood-based industries [5]. Several studies concerning oil palm wood have been conducted, such as the use of trunks in plywood [5], sawn timber [6], laminated wood [7], particleboard [8], and bio-composites [9,10]. However, one interesting aspect has not been touched on, namely, the properties of oil palm wood as a thermal insulation material in buildings. Determining these properties is essential to producing bio-insulation materials with good thermal resistance and applying them to conventional buildings.

Thermal insulation materials play an important role in energy conservation by slowing the heat transfer rate [11]. Various innovations in natural-fiber-based thermal insulation materials have been developed. Binici et al. [12] developed a thermal insulation material from corn cobs with epoxy resin. The findings of this study have the potential to achieve two industrial aims: Creating new functional construction materials and reducing the environmental impact of agricultural waste. Furthermore, insulating materials from vegetable farm waste, such as rye straw, barley straw, wheat straw, oat straw, rice straw, flax boon, and rice husk, have been reported [13]. Manohar [14] measured the thermal conductivity of palm, coconut, and sugarcane fibers as building insulation materials. The results showed that the coefficient of thermal conductivity ranged from 0.055 to 0.091 W/mK. Another study reported the thermal properties of structural materials made of 40% palm bunch fibers and phenol-formaldehyde [15]. The thermal conductivity value of the developed sample was 0.293 W/mK. In addition, some researchers have reported the valorization of tree bark fiber for thermal insulation panels, such as spruce, black locust, larch, and poplar bark [16,17,18,19,20,21]. These investigations showed that bark residue is a potential alternative raw material for efficient bio-based thermal insulation.

Generally, thermosetting polymers such as urea-formaldehyde (UF), phenol-formaldehyde (PF), and epoxy are used as a binder to form thermal insulation panels. This resin type is known to have good mechanical properties but is not environmentally friendly because of its poor biodegradability [22]. Besides that, the effect of harmful exposure to formaldehyde in the air can cause skin irritation, watery eyes, nausea, and coughing. If prolonged exposure occurs, it can cause cancer, such as nasopharyngeal cancer in humans [23]. Therefore, it is important to promote biopolymers from natural resins to replace synthetic resins in order to conserve the environment. Previous researchers have studied natural-fiber-reinforced insulating materials and bio-binders from cassava starch, bone glue, and corn starch, including sugar palm fiber [24], bamboo powder [23], and date palm tree surface fibers [25]. Tapioca starch is one candidate for binders to make completely natural wood-based panels, such as insulating materials. In addition, many research groups have studied bio-insulating materials based on natural fibers and bio-adhesive agents from tannins [26], soybean protein [27], starch [28], and lignin [29]. Manfred et al. [30] and Antov et al. [31] reviewed bio-based adhesives for wood composites. They reported that bio-based adhesives of lignin, starch, and tannins can potentially be used to produce eco-friendly wood composite materials. Furthermore, some researchers have extensively studied the potential utilization of lignosulfonates to produce eco-friendly particleboard [32,33,34]. Several research groups have used tapioca starch as a bio-binder in bio-insulation materials, including ramie fiber [35], sugarcane bagasse [36], water hyacinth [37], bamboo [25,38], and oil palm empty fruit bunches (OPEFB) [39].

Thermal insulation materials require thermal resistance and good mechanical and physical properties. Previous studies verified the hybridization process as one technique that can be used to improve the mechanical properties and durability of materials. Board made from mixed OPEFB and oil palm wood showed improved mechanical properties [40]. Other findings showed that OPEFB hybridization with sugarcane increased the tensile strength of the bio-composite by up to 18.8% [41]. Furthermore, a jute hybrid composite showed less water absorption than pure OPEFB oil palm composite [42].

The objective of this study was to manufacture thermal bio-insulation materials based on oil palm wood and investigate the effect of hybridization and particle size on the properties of the panels. We produced hybrid oil palm wood/ramie fiber bio-insulation materials with tapioca starch as the bio-binder.

## 2. Materials and Methods

### 2.1. Material Preparation

Oil palm trunks (*Elaeis guineensis*) aged about 25 years were cut and collected from plantations in Aceh. Herein, we used the inner oil palm trunk as oil palm wood. Meanwhile, ramie fibers *(Boehmeria nivea)* were obtained from plantations in Yogyakarta, Indonesia. Tapioca starch was used as a bio-binder resin and purchased at the local market. Tapioca starch is composed of amylose and amylopectin, both of which play a role in binding. Amylopectin acts as an adhesive, while amylose acts as a hardener [43]. Table 1 shows the chemical composition and the mechanical and physical properties of OPW, ramie fiber, and tapioca starch.

The OPW was cut to dimensions 200 mm × 50 mm × 50 mm, dried, crushed, and pulverized into particles using a ball mill (Planetary Mill, Fritsch, Germany). Particles were grouped into three fineness levels, namely, coarse (passed 0.84 mm), medium (passed 0.42 mm), and fine (passed 0.07 mm). Meanwhile, ramie fibers were cut into short fibers of length 1–5 mm. Figure 1 shows the transformation of the raw OPW material and ramie fibers.

### 2.2. Treatment of Fiber

OPW particles were boiled in water at 100 °C for 30 min. According to Jumhuri et al. [45], pre-treatment of oil palm wood particles by soaking in hot water for 30 min improved the mechanical properties of the resulting particleboard. The ramie fibers were immersed in 5% NaOH solution for 1 h before being washed to a neutral pH. Both materials were then dried in an oven at 80 °C for 24 h to 10–15% moisture content.

### 2.3. Manufacturing of Bio-Panels

The panels were manufactured according to the formulations exhibited in Table 2. The mats were manufactured using a laboratory-type hydraulic hot press (Hot-Pressing HM100, Hikmah Machine, Indonesia) with a pressure capacity of 100 MPa. Fibers as reinforcement were mixed with tapioca starch, then 100 mL of hot water was sprayed using a spray gun. Then, the mats were stirred using a mixer (model HM-620, Miyako, Jakarta Barat, Indonesia) for 5 min until thoroughly mixed. Then, the mats were spread into a mold with a size of 150 mm × 150 mm × 30 mm. The mats were pressed at a temperature of 150 °C and pressure of 9.8 MPa for 15 min, to a thickness of 10 mm, with a target density of 0.7 g/cm^3^. In this work, the panels were formed with two-stage pressing, pre-pressed at a pressure of 9.8 MPa for 5 min before being hot pressed for 10 min at 150 °C under 9.8 MPa. This condition followed a previous study that used a hot full press for 10 min [45,46]. The elevated long-pressing time activates the starch so that the binder forms and makes it possible to evaporate the water present in the adhesive composition [33,47,48]. The manufactured panels were conditioned in a conditioning room maintained at 25 °C and 65% relative humidity for seven days. Figure 2 shows the physical condition of the bio-panels.

### 2.4. Physical Measurements

This study included three physical tests: Density, water absorption, and dimensional stability. Three samples from each bio-panel of dimensions 50 mm × 50 mm × 10 mm were made for the density, water absorption, and thickness swelling measurements. The physical properties of the panels were determined and evaluated according to the SNI 03-2105-2006 standard [49].

### 2.5. Mechanical Measurements

Five bending strength samples were prepared from each bio-panel. In the present work, an MTS EXCEED Model E43 universal testing machine with a crosshead speed of 2 mm/min was used for the bending tests. The bending tests were carried out according to ASTM D790-30 [50], and the specimen dimensions were 130 mm × 20 mm × 10 mm.

### 2.6. Thermal Measurements

Thermogravimetric analysis (TGA) was used to determine the thermal degradation characteristics of the bio-panels. For the tests, we used a SHIMADZU DTG 60 Thermal Analyzer following the ASTM E1131-08 standard. A sample weighing about 5 mg was heated from 30 °C to 600 °C under a nitrogen atmosphere at a 20 mL/min flow rate and a temperature rate of 40 °C/min. In addition, the thermal conductivity of samples was measured using an insulated box (PHYWE SYSTEME GMBH 37070 Göttingen, Germany). For the test method, we referred to ASTM C177-97 [51] under steady-state temperature conditions. Temperature measurements were carried out using thermocouples located inside and outside the box and the walls of the specimen. The measurement data were tabulated and calculated using the Fourier Equation (1) to obtain the thermal conductivity value [52]:(1)$$Q_{\mathrm{con}.}=-\mathrm{kA}\frac{\mathrm{dt}}{\mathrm{dx}}\;\left(W\right)$$
where k is the thermal conductivity, A is the wall area, and (dt/dx) is constant.

## 3. Results and Discussion

### 3.1. Physical Properties of the Bio-Panels

The physical properties are important parameters in determining the quality of wood-based panels. Table 3 shows that the moisture content of the bio-panels ranged from 13% to 14%, and the density was between 0.66 and 0.79 g/cm^3^. The moisture contents of all samples with different particle sizes were comparable, while the densities were different.

The results showed that panels made from fine particles had a higher density than did those made from coarse particles. Sample H3 had the highest density, followed by H2 and H1, while the density of the pure bio-panels was ordered as follows: P1 < P2 < P3. The use of smaller particles in the panels results in better compaction of the mat, whereas larger particles may result in more and larger pores [53]. Previous studies have reported that boards from Washingtonia with particles of size 0.25 to 1 mm have a higher density than do those with particles of size 2 to 4 mm [54]. In addition, hybrid panels composed of oil palm wood and ramie fiber showed a higher density than did panels made solely using oil palm wood. This effect may be due to the raw material density affecting the density of the panels. The density of oil palm wood is 0.15–0.4 g/cm^3^, and that of ramie fiber is 1.5 g/cm^3^. Previous studies showed similar results for OPEFB hybrid panels and oil palm trunks [40].

The water resistance of a panel is an important parameter, especially when natural fiber is used in its construction. The water absorption obtained for all samples was in the range from 54.76% to 65.48% (Table 3). Sample P1 absorbed the greatest amount of water (65.48%) among the pure panels, while H1 showed the same tendency (65.12%). Both samples were made of coarse particles and of low density. Lamaming et al. [55] reported that particleboard made from old palm oil wood with a particle size of 500–1000 µm had higher water absorption capacity (211.03%) compared to that with a particle size range of 100–500 µm (90.70%). Oil palm wood panels have a high water uptake capacity due to the presence of more hydroxyl groups in the parenchyma tissue, which allows for more hydrogen bonding.

Furthermore, the parenchyma acts like a sponge, allowing the oil palm wood particles to absorb water more easily [6]. However, hybridization with ramie fiber slightly reduces water absorption. This study proved a 3–5% reduction when hybridizing ramie fiber and oil palm wood. This result is supported by [56], which stated that the addition of alkaline-treated ramie fiber can improve the water-resistance performance. According to [41], the addition of co-fiber has the potential to reduce water absorption due to the compatible characteristics of the mixed material, which reduces the voids and porosity of the exposed surface area on composites, thereby reducing water absorption.

The dimensional stability of panels followed a similar trend to the water absorption properties (Table 3). The value of the dimensional stability of the samples ranged from 18.18% to 30.28%. The bio-panel P1 showed the poorest rate of dimensional stability (30.28%) among all the panels. In contrast, the hybrid panel H3 was the best, with a value of 18.18%. In other words, the panel showed less thickness swelling. According to [42], a panel structure with higher density and mixed co-reinforcement leads to an increased dimensional stability rate. This finding is better than those for eucalyptus particleboard with methane diphenyl diisocyanate polymer adhesive and UF as an adhesive [57] and sago particleboards reinforced with PF and UF binder [58].

Table 3 summarizes all the properties that were evaluated, where hybrid panels were found to be better than pure panels. The density variation of 20% of the hybrid panel (0.66 to 0.79 g/cm^3^) affected all the panel properties, even for those of the same composition. This is due to the effect of the particle size difference: Improved particle size tends to increase the panels’ thermal resistance and mechanical performance.

### 3.2. Correlation of Density with the Thermal Conductivity of Bio-Panels

Thermal conductivity is an important property of thermal insulation materials, and the effectiveness is different for each material. Figure 3 depicts the thermal conductivity of the bio-panels specified in Table 2. The produced panels exhibited medium density that ranged from 0.66 g/cm^3^ to 0.79 g/cm^3^, and their thermal conductivity was between 0.067 W/mK and 0.154 W/mK. Samples H1 and P3 exhibited the lowest and highest thermal conductivity (0.067 W/mK and 0.154 W/mK), respectively, whereas samples H1 and H3 showed the lowest and highest density (0.66 g/cm^3^ and 0.79 g/cm^3^), respectively. For all conditions, both types of bio-panels had nearly the same density. Nevertheless, the thermal conductivity of the hybrid was slightly lower than that of the pure panel. These results show that the ramie fiber hybridization provides better thermal conductivity when used with oil palm wood particles. A similar tendency was also reported in a previous study [59,60].

The thermal conductivity of panels is related to their density. A higher panel density is correlated with higher thermal conductivity. As the panel density increases, the solids are increased, and voids are decreased. According to [61], voids act as scattering centers for phonons and absorb a small portion of the heat conduction volume of the material, resulting in lower thermal conductivity. The correlation between density and thermal conductivity of the produced bio-panels and a comparison with various insulation materials is given in Figure 3. Overall, there is a direct correlation between the density and thermal conductivity of all boards. The thermal conductivity decreases as the sample density decreases. Some authors found such a relationship in previous studies, while others did not fully achieve a relationship between these properties [23,62]. 

The thermal conductivity of the produced panels is comparable with that of sunflower (0.06 W/mK) [63], date palm (0.07 W/mK) [25], wheat straw (0.56 W/mK) [13], wood fiber/Ecovio (0.11 W/mK) [64], banana/PP (0.157 W/mK) [65], and bamboo/bone (0.118 W/mK) [23]. Therefore, some samples (P1, H1, and H2) could be deemed potential thermal insulation materials because their thermal conductivity is less than 0.1 W/mK [63].

### 3.3. Bending Strength and Thermal Conductivity

In addition to thermal performance, evaluation of the mechanical parameters is very important to measure the performance of bio-panels under various types of stress during their application. The bending strength of bio-panels ranged from 11.49 MPa to 18.15 MPa, and the modulus of elasticity ranged between 1864 MPa and 3093 MPa (given in Table 3). Bio-panels H1 and P3 showed the highest and lowest bending strength, respectively, while panels P1 and P3 exhibited the highest and lowest modulus of elasticity, respectively.

Overall, the hybrid panels had higher bending strength than the pure panels (Figure 4). The hybridization of pure oil palm wood with ramie fiber improved the bending strength of the panels. The average bending strength of the pure bio-panels was 25% lower than that of the hybrids. This was probably influenced by the single fiber and the low mechanical properties of the OPW raw material. This result was confirmed by [40], who stated that the structure of a bio-panel with a single fiber and the deterioration of the compact structure is responsible for a decrease in mechanical properties.

In addition, the bending strength also depends on the particle size. As the size of the particles increased, the bending strength increased. Previous research on the particle geometry and bending strength of panels concluded that larger particle contact areas resulted in higher strength properties [66,67]. The viscoelastic nature of carbohydrate-rich oil palm trunk particles is one of the main reasons for the acceptable mechanical properties of the samples [68]. According to [55,69], the particle size in bio-panels affects board characteristics such as mechanical strength, water absorption, thickness expansion, surface roughness, and linear stability.

Figure 4 depicts how the thermal conductivity is related to the bending strength of the bio-panels. Panel H1 and P3 showed the lowest and highest thermal conductivity and highest and lowest bending strength, respectively. Overall, hybrid samples have better thermal resistance and bending strength than pure panels. This is also related to particle size and density. As the particle size increases, it is accompanied by a decrease in the density and a decrease in the thermal conductivity coefficient of the panels.

Several previous researchers have reported a similar conclusion. Alabdulkarem et al. [59] reported that hybrid bio-panels made with agave and apple of Sodom fibers produce better thermal resistance than panels with single fibers. The thermal conductivity of jute-fiber-reinforced epoxy composites was 12% more than that of EFB/jute hybrid composites [70].

### 3.4. TGA and DTGA Spectra

Figure 5 shows the results of thermogravimetric and derivative analysis of the bio-panels. Both showed a similar degradation pattern, where the filler decomposed in three steps during the heating process. Initial degradation in the range 35–93 °C caused weight loss of 7.1% and 5.4% for pure and hybrid bio-panels, respectively. This was due to water evaporation from the samples [25,71,72].

Furthermore, major degradation happened between 306 °C and 411 °C with a weight loss of about 63.91% for pure bio-panels (Figure 5a) and between 310 °C and 408 °C with a weight loss of 56.88% for hybrid bio-panels (Figure 5b). This weight loss was associated with the degradation of fiber, carbon dioxide, water, and decomposition of tapioca starch [37,73]. In the last stage, above 408 °C, the weight loss for the hybrid bio-panels (3.91%) was lower than that for the pure bio-panels (7.06%). This phenomenon was found in other studies [72,74]. Table 4 shows the temperatures associated with weight loss of 25%, 50%, and 75% from the bio-panels. This result is almost identical to the results obtained by [25], which showed a 50% loss of date fiber mass at 364 °C.

Figure 5 also shows the derivative thermogravimetric analysis (DTGA) curve, which indicated the occurrence of major degradation. In this curve, the main degradation occurred at 306 °C and 339 °C for pure bio-panels and at 310 °C and 341 °C for hybrid bio-panels. This finding is better than the DTGA findings for a bio-panel made from oil palm trunk with ammonium dihydrogen phosphate at 200 °C and 330 °C [75]. Different fiber types, fiber treatment processes, and matrices all affect the thermal stability behavior of natural-fiber-reinforced panels [76].

## 4. Conclusions

Thermal bio-insulation panels based on oil palm wood hybridized with tapioca starch binders were successfully produced using a hot press. In general, the results show that ramie fiber hybridization improved the physical, mechanical, and thermal properties of the bio-panels. Furthermore, the particle size also affected the properties of the bio-panels. The density, water resistance, and dimension stability of the bio-panels increased with increasing particle size. However, this was contrary to the results regarding bending strength and thermal conductivity. The bending properties of the hybrid panels were better than those of pure oil palm wood. The thermal conductivity of oil palm wood panels ranged from 0.067 to 0.154 W/mK, following their density. The thermal conductivity of the panels with larger particles was within a lower range than that of panels with the highest compactness. The panels P1, H1, and H2 produced thermal conductivity values less than 0.1 W/mK. Hence, they can be used as thermal insulation materials. Thermogravimetric analysis revealed that the hybrids’ thermal stability was better than that of pure bio-panels. Major degradation occurred at temperatures of 306–411 °C for pure and 310–408 °C for hybrid bio-panels. Finally, this study shows that waste oil palm wood, which causes environmental pollution, can be utilized to manufacture commercially viable and satisfactory thermal insulation materials.

## Figures and Tables

**Figure 1 polymers-13-03287-f001:**
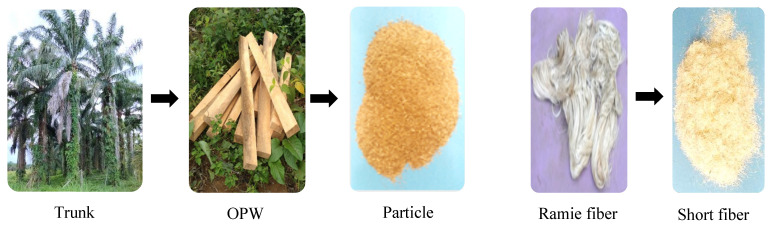
Raw material transformation of OPT to OPW to particles, and ramie fiber.

**Figure 2 polymers-13-03287-f002:**
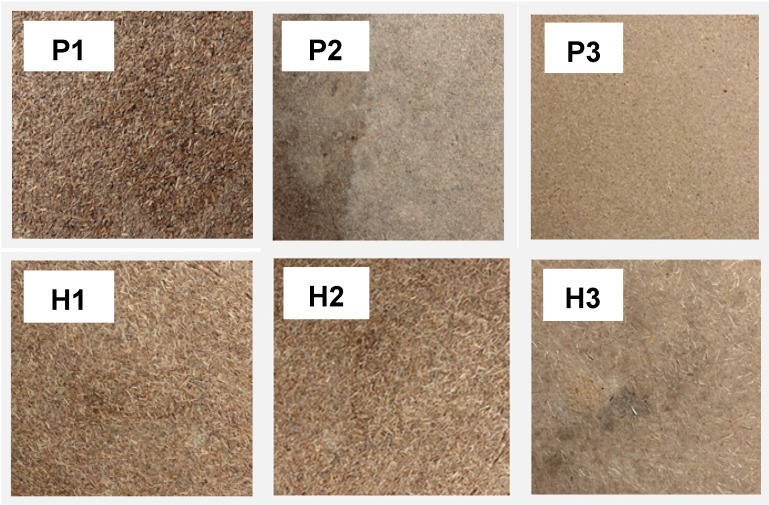
The physical condition of the bio-panels.

**Figure 3 polymers-13-03287-f003:**
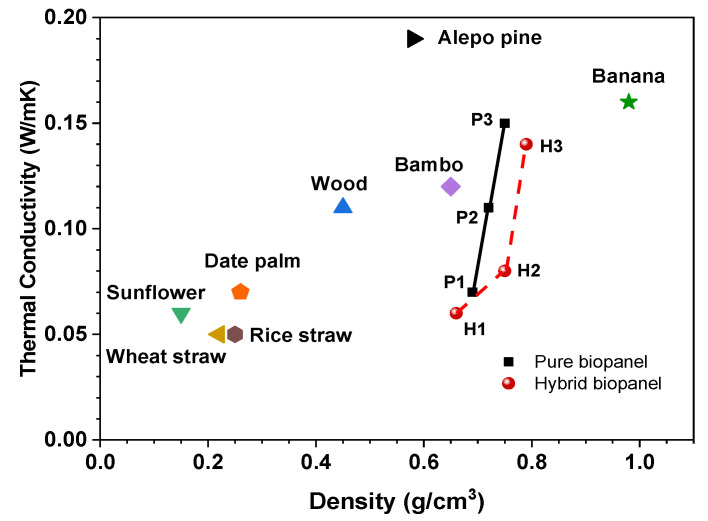
Correlation between the thermal conductivity and density of various insulation materials.

**Figure 4 polymers-13-03287-f004:**
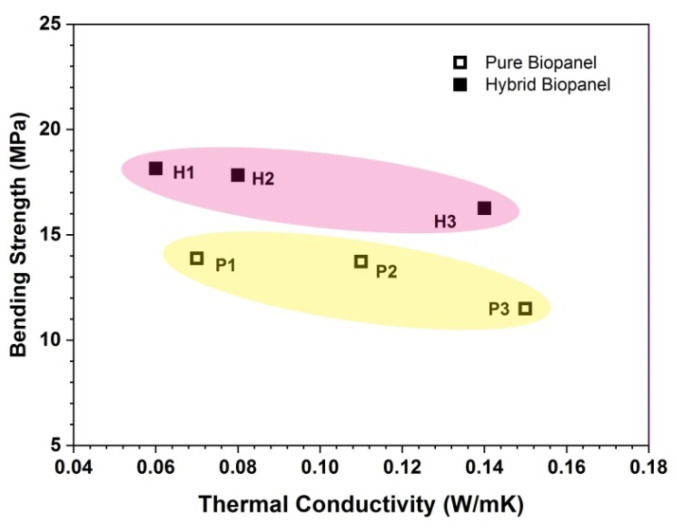
Relationship of bending strength with the bio-panels’ thermal conductivity.

**Figure 5 polymers-13-03287-f005:**
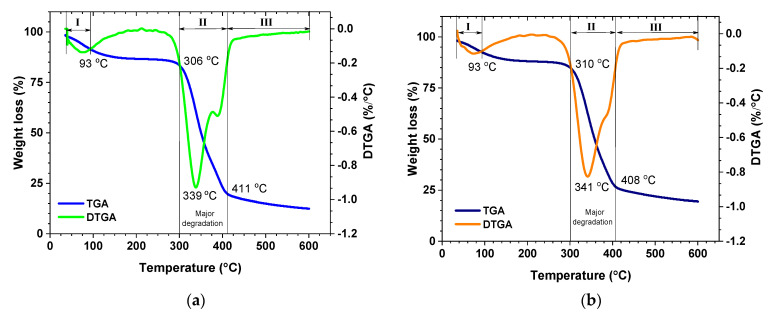
TGA and DTGA spectra of bio-panels: (**a**) Pure; (**b**) hybrid.

**Table 1 polymers-13-03287-t001:** Chemical composition and mechanical and physical properties of the hybrid bio-panel forming materials [36,40,44].

Description	OPW	Ramie Fiber	Tapioca Starch
Chemical constituents (%)	29–37	68.6–76.2	
Cellulose	12–17	13.1–16.7	
Hemi cellulose	18–23	0.60–0.80	-
Lignin	-	-	-
Amylose	-	-	-
Amylopectin	-	-	-
Physical and mechanical properties	-	-	17
Density (g/cm3)	0.15–0.4	1.50	83
Tensile strength (MPa)	300–600	290–1060	
Young’s modulus (GPa)	15–32	5–128	

**Table 2 polymers-13-03287-t002:** Formulations of bio-panels.

Type	Code	Particle Classification	OPW (%)	Ramie Fiber (%)	Tapioca Starch (%)
Pure bio-panels	P1	coarse	70	0	30
	P2	medium	70	0	30
	P3	fine	70	0	30
Hybrid bio-panels	H1	coarse	50	20	30
	H2	medium	50	20	30
	H3	fine	50	20	30

**Table 3 polymers-13-03287-t003:** Properties of the bio-panels.

Kode	Density (g/cm^3^)	MC (%)	WA (%)	DS (%)	BS (MPa)	MOE (MPa)	k (W/mK)
Pure bio-panels							
P1	0.69	13	65.48	30.28	13.88	3093	0.071
P2	0.72	13	58.72	29.70	13.73	2604	0.110
P3	0.75	13	56.69	24.49	11.49	1864	0.154
Hybrid bio-panels							
H1	0.66	14	65.12	29.63	18.15	2605	0.067
H2	0.75	13	55.59	25.00	17.83	2546	0.089
H3	0.79	13	54.75	18.18	16.26	1984	0.148

MC: Moisture Content; WA: Water Absorption; DS: Dimensional Stability; BS: Bending Strength; MOE: Modulus of Elasticity; k: Thermal Conductivity.

**Table 4 polymers-13-03287-t004:** Temperatures of weight loss and residual mass.

Bio-Panel	Weight Loss at Temp. Decomposition (°C)			Weight Residue at 600 °C
	25%	50%	75%	
**Pure**	321	349	394	12
**Hybrid**	327	358	421	19

## Data Availability

Not applicable.

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
