# Peer review of "Characterization of Thermal Bio-Insulation Materials Based on Oil Palm Wood: The Effect of Hybridization and Particle Size"

_polymers, 2021, doi:10.3390/polym13193287_

Round 1
Reviewer 1 Report
In the Abstract
In my opinion, instead of "modulus bending", "Modulus of Elasticity in Bending" should be used, or just "Modulus of Elasticity". I recommend for this property also use the dimension MPa (as for the bending strength) and not GPa.
Please add the main novelty in your research in this section.
In the Introduction:
Please expand the review for bio-based binders, not only for starch but also for tannins, lignin, lignosulfonates and others. On this basis, increase the number of references.
Please specify the sentence “Bio-binders are used as resins to make completely natural insulating materials” (lines 88 ÷ 89). Bio-based binders are used not only for insulation materials but also for wood-based panels for furniture production, construction purposes and so on.
Please justify the regime used for the pre-treatment of fibres (lines 136-140).
Why were particles boiled in water at 100O C for 30 minutes and not for example for 40 minutes? Please provide references justifying the regimes you used.
The same applies to a much greater extent to the hot-pressing regime. The specified pressure of 9.8 MPa is very high. Usually, the first-stage pressure for panels with a density of 0.70 g/cm3 is in the range of 4 ÷ 5 MPa. You may have given the manometric pressure (the pressure in the hydraulic system). If this is the case, please give the specific pressure (pressure on the pressed material). Please provide in more detail the regime of hot-pressing - how many stages are used, pressure and time at each stage. How do you choose the press factor (time of hot-pressing)?
Please explain how you set the target density of the panels. Typically, the density of insulation materials is in the order of less than 0.4 g / cm3.
In the Results and Conclusions
Please explain the variation in the density from 0.66 to 0.79 g / cm3. This variation is 20% and it will affect all the properties of the panels, even if they are of the same composition.
Author Response
Response to Reviewers 1
Dear Polymers Editorial and my honourable reviewers
I am honoured to receive your comprehensive and well-thought comments regarding my recent submission. I am therefore writing to provide some clarity on the subject raised and try to fulfil your request on addressing the highlighted comments. For your reviewing convenience, the additional information which will be incorporated within the revised manuscript. Plus I have assigned the comment and questions by the reviewer with the letter Q and my corresponding response as R.
|
|
|
In the Abstract |
|
Q1 |
: |
In my opinion, instead of "modulus bending", "Modulus of Elasticity in Bending" should be used, or just "Modulus of Elasticity". I recommend for this property also use the dimension MPa (as for the bending strength) and not GPa |
|
R1 |
: |
I appreciate your advice. I have revised the sentence in the manuscript. |
|
|
|
|
|
Q2 |
: |
Please add the main novelty in your research in this section |
|
R2 |
: |
I appreciate your suggestion. I have added the novelty of this paper in the abstract of the manuscript on lines 15-19. |
|
|
|
|
|
|
|
In the Introduction: |
|
Q3 |
: |
Please expand the review for bio-based binders, not only for starch but also for tannins, lignin, lignosulfonates and others. On this basis, increase the number of references |
|
R3 |
: |
I appreciate your suggestion. I have added the related reference and include references in the manuscript in lines 106-114. |
|
|
|
|
|
Q4 |
: |
Please specify the sentence “Bio-binders are used as resins to make completely natural insulating materials” (lines 88 ÷ 89). Bio-based binders are used not only for insulation materials but also for wood-based panels for furniture production, construction purposes and so on |
|
R4 |
: |
I appreciate your suggestion. I have revised the statement in the manuscript in lines 104-106. |
|
|
|
|
|
Q5 |
: |
Please justify the regime used for the pre-treatment of fibres (lines 136-140). |
|
R5 |
: |
I appreciate your suggestion. I have justified the regime used for the pre-treatment of fibers in lines 165-171 |
|
|
|
|
|
Q6 |
: |
Why were particles boiled in water at 100O C for 30 minutes and not for example for 40 minutes? Please provide references justifying the regimes you used |
|
R6 |
: |
I have appreciated your question. I have explained the pretreatment of particles boiled in water at 100OC for 30 minutes and added references justifying the regimes used in the manuscript (lines 165-171). |
|
|
|
|
|
Q7 |
: |
The same applies to a much greater extent to the hot-pressing regime. The specified pressure of 9.8 MPa is very high. Usually, the first-stage pressure for panels with a density of 0.70 g/cm3 is in the range of 4 ÷ 5 MPa. You may have given the manometric pressure (the pressure in the hydraulic system). If this is the case, please give the specific pressure (pressure on the pressed material). Please provide in more detail the regime of hot-pressing - how many stages are used, pressure and time at each stage. How do you choose the press factor (time of hot-pressing)? |
|
R7 |
: |
Thank you for your comment and suggestion. Your suggestion could be very relevant to improve our validation method in future work. In this study, the panels were manufactured based on oil palm wood (OPW), which the essential characteristics of OPW have low physical and mechanical properties. In previous work, we evaluated pressures below 9.8 MPa we did not achieve the target density of 0.7 g/cm3. Therefore, we increased the pressure to 9.8 MPa to get the density target.
The choose the hot-pressing factor, many stages are used, pressure and time at each stage after explaining in the manuscript (lines 182-185) |
|
|
|
|
|
Q8 |
: |
Please explain how you set the target density of the panels. Typically, the density of insulation materials is in the order of less than 0.4 g / cm3. |
|
R8 |
: |
Thank you for your comment, and we agree with your statement that the density of insulation materials is typically in the order of less than 0.4 g/cm3. However, in previous works, we found that the density of OPW panels in the range of 0.4-0.5 g/cm3 has low physical and mechanical properties. Therefore, we increased the target density to 0.7 g/cm3 because this work is also an effort to obtain panels with low thermal conductivity and good physical and mechanical properties. |
|
|
|
|
|
|
|
In the Results and Discussion |
|
Q9 |
: |
Please explain the variation in the density from 0.66 to 0.79 g / cm3. This variation is 20% and it will affect all the properties of the panels, even if they are of the same composition. |
|
R9 |
: |
I appreciate your suggestion. I have explained these in the manuscript in lines 290-295. |
|
|
|
|

Reviewer 2 Report
The manuscript is focused on investigation and evaluation of the physical, mechanical, and thermal properties of bio-based insulation materials, fabricated from oil palm wood and ramie, bonded with tapioca starch as a bio-adhesive.
The title of the manuscript (lines 2-4) is clear and informative.
In general, the abstract (lines 15-32) and the keywords (lines 33-34) are specific and correspond to the title, aims and objectives of the manuscript.
In lines 19-20, the sentence “A panel was prepared using hot-press at a temperature and pressure constant.” is not very clear, and should be revised, e.g. by providing specific information on the press regime. In addition, I believe the authors produced more than just one panel, please revise accordingly.
In line 30, please delete the unnecessary “was”.
In lines 33-34, I’d suggest to replace “natural binder” with “bio-based adhesive”, and “physical and bending properties” with “physical and mechanical properties”.
In line 38, please revise the first sentence, it’s not very clear, e.g. “Indonesia is the biggest producer of palm oil worldwide”.
In line 71, I’d suggest the authors to add some information related to the valorisation of tree bark for thermal insulation panels. Please refer to the following recent references in the field:
https://doi.org/10.1007/s00107-019-01436-5
https://doi.org/10.1016/j.conbuildmat.2017.04.204
https://doi.org/10.3390/polym13142287
https://doi.org/10.3390/polym13111799
https://doi.org/10.1016/j.conbuildmat.2020.121577
https://doi.org/10.3390/app10165594
In lines 79-80, there is no need to use capital letters for urea-formaldehyde and phenol-formaldehyde resins, please revise.
In line 83, apart from the poor biodegradability, the most important drawback of the commercial thermosetting resins such as UF and PF is the emission of VOCs, including free formaldehyde from the finished composites, which are linked with serious negative effects on human health, including cancer. Please add some information and references.
In lines 86-88, I’d recommend to the authors to add some information on the other important types of bio-based adhesives for production of lignocellulosic composites, such as tannins, soy, and lignin. Please refer to the following interesting publications:
https://doi.org/10.7569/RAA.2021.097307
https://doi.org/10.3390/polym12051115
doi:10.32604/jrm.2021.012782
doi.org/10.37763/wr.1336-4561/65.1.051062
https://doi.org/10.1007/s00226-020-01248-4
https://doi.org/10.3390/ma14174875
https://doi.org/10.3390/polym13162775
https://doi.org/10.1007/s00107-006-0130-z
https://doi.org/10.1016/j.jclepro.2021.127892
http://doi.org/10.1007/s00226-017-0957-y
http://doi.org/10.3390/molecules26154526
In lines 107-109, the sentence “Different analysis equipment, such as universal bending testing machine, high insulation house, and TGA, are used.” does not belong to the Introduction, please remove it.
Overall, the Introduction part is well written and informative, but can be further elaborated based on the comments provided.
In lines 112 and 114, I’d suggest to add the botanical names of the species used in the research, i.e. oil palm and ramie.
In lines 115-116, please add some information and characteristics of the bio-based binder (tapioca starch).
In line 119, Table 1 is not the correct one. It is equal to Table 2 (line 150). Please add the correct table about the chemical composition and properties of the materials used in the research.
In line 143, please add some information about the hot press used (company producer, city, country).
In line 145, please add information about the mixer used.
In line 148, please explain the selected hot pressing parameters, e.g. based on preliminary trials, other references, etc.
In line 166, please add the respective standard (SNI 03-2105-2006) in the manuscript references. The same comment applies to line 171 (ASTM D70), and line 181 (ASTM C177-9). In addition, I believe the correct standard in line 181 should be “ASTM C177 – 19. Standard Test Method for Steady-State Heat Flux Measurements and Thermal Transmission Properties by Means of the Guarded-Hot-Plate Apparatus”, please revise.
In general, the Results and Discussion section is well-structured, concise, informative and discussed with previously published research works.
The Conclusions (lines 353-373) are consistent with the results and reflect the main findings of the study.
The references cited are appropriate and correspond to the topic of the manuscript. The inclusion of additional references in all sections of the manuscript is highly recommended. This will significantly increase the value of the presented work.
Best regards!
Author Response
Response to Reviewers 2
Dear Polymers Editorial and my honourable reviewers
I am honoured to receive your comprehensive and well-thought comments regarding my recent submission. I am therefore writing to provide some clarity on the subject raised and try to fulfil your request on addressing the highlighted comments. For your reviewing convenience, the additional information which will be incorporated within the revised manuscript. Plus I have assigned the comment and questions by the reviewer with the letter Q and my corresponding response as R.
|
Q1 |
: |
The manuscript is focused on investigation and evaluation of the physical, mechanical, and thermal properties of bio-based insulation materials, fabricated from oil palm wood and ramie, bonded with tapioca starch as a bio-adhesive |
|
R1 |
: |
We would like to thank the reviewers for this statement |
|
|
|
|
|
Q2 |
: |
The title of the manuscript (lines 2-4) is clear and informative. |
|
R2 |
: |
We would like to thank the reviewers for this statement |
|
|
|
|
|
Q3 |
: |
In general, the abstract (lines 15-32) and the keywords (lines 33-34) are specific and correspond to the title, aims and objectives of the manuscript. |
|
R3 |
: |
We would like to thank the reviewers for this statement |
|
|
|
|
|
Q4 |
: |
In lines 19-20, the sentence “A panel was prepared using hot-press at a temperature and pressure constant.” is not very clear, and should be revised, e.g. by providing specific information on the press regime. In addition, I believe the authors produced more than just one panel, please revise accordingly |
|
R4 |
: |
I appreciate your reminding. I have revised the sentence in manuscript (lines 23-24) |
|
|
|
|
|
Q5 |
: |
In line 30, please delete the unnecessary “was”. |
|
R5 |
: |
I appreciate your reminding. I have deleted the character. |
|
|
|
|
|
Q6 |
: |
In lines 33-34, I’d suggest to replace “natural binder” with “bio-based adhesive”, and “physical and bending properties” with “physical and mechanical properties”. |
|
R6 |
: |
I appreciate your suggestion. I have replaced this word (lines 40-41). |
|
|
|
|
|
Q7 |
: |
In line 38, please revise the first sentence, it’s not very clear, e.g. “Indonesia is the biggest producer of palm oil worldwide”. |
|
R7 |
: |
I appreciate your suggestion. I have the replace the sentence in manuscript (lines 44-48) |
|
|
|
|
|
Q8 |
: |
In line 71, I’d suggest the authors to add some information related to the valorisation of tree bark for thermal insulation panels. Please refer to the following recent references in the field: https://doi.org/10.1007/s00107-019-01436-5 https://doi.org/10.1016/j.conbuildmat.2017.04.204 https://doi.org/10.3390/polym13142287 https://doi.org/10.3390/polym13111799 https://doi.org/10.1016/j.conbuildmat.2020.121577 https://doi.org/10.3390/app10165594 |
|
R8 |
: |
I appreciate your suggestion. I have included the reference in to the manuscript (lines 87-91) |
|
|
|
|
|
Q9 |
: |
In lines 79-80, there is no need to use capital letters for urea-formaldehyde and phenol-formaldehyde resins, please revise. |
|
R9 |
: |
I have replaced the capital letters with a small case |
|
|
|
|
|
Q10 |
: |
In line 83, apart from the poor biodegradability, the most important drawback of the commercial thermosetting resins such as UF and PF is the emission of VOCs, including free formaldehyde from the finished composites, which are linked with serious negative effects on human health, including cancer. Please add some information and references. |
|
R10 |
: |
I appreciate your suggestion. I have the revised the sentence in manuscript (lines 96-99) |
|
|
|
|
|
Q11 |
: |
In lines 86-88, I’d recommend to the authors to add some information on the other important types of bio-based adhesives for production of lignocellulosic composites, such as tannins, soy, and lignin. Please refer to the following interesting publications: https://doi.org/10.7569/RAA.2021.097307 https://doi.org/10.3390/polym12051115 doi:10.32604/jrm.2021.012782 doi.org/10.37763/wr.1336-4561/65.1.051062 https://doi.org/10.1007/s00226-020-01248-4 https://doi.org/10.3390/ma14174875 https://doi.org/10.3390/polym13162775 https://doi.org/10.1007/s00107-006-0130-z https://doi.org/10.1016/j.jclepro.2021.127892 http://doi.org/10.1007/s00226-017-0957-y http://doi.org/10.3390/molecules26154526 |
|
R11 |
: |
I appreciate your suggestion. I have added the related reference and include references in the manuscript (lines 106-114) |
|
|
|
|
|
Q12 |
: |
In lines 107-109, the sentence “Different analysis equipment, such as universal bending testing machine, high insulation house, and TGA, are used.” does not belong to the Introduction, please remove it. |
|
R12 |
: |
I appreciate your suggestion. I have deleted the sentence |
|
|
|
|
|
Q13 |
: |
Overall, the Introduction part is well written and informative, but can be further elaborated based on the comments provided. |
|
R13 |
: |
We would like to thank the reviewers for this statement |
|
|
|
|
|
Q14 |
: |
In lines 112 and 114, I’d suggest to add the botanical names of the species used in the research, i.e. oil palm and ramie. |
|
R14 |
: |
I have add the botanical names of oil palm (Elaeis guineensis) and ramie (Boehmeria nivea) in the manuscript. |
|
|
|
|
|
Q15 |
: |
In lines 115-116, please add some information and characteristics of the bio-based binder (tapioca starch). |
|
R15 |
: |
I appreciate your suggestion. I have the added some information and characteristics in manuscript (lines 138-142) |
|
|
|
|
|
Q16 |
: |
In line 119, Table 1 is not the correct one. It is equal to Table 2 (line 150). Please add the correct table about the chemical composition and properties of the materials used in the research. |
|
R16 |
: |
I appreciate your reminding. I have corrected the table with the real one |
|
|
|
|
|
Q17 |
: |
In line 143, please add some information about the hot press used (company producer, city, country). |
|
R17 |
: |
I appreciate your suggestion. I have the added some information about the hot press used in the manuscript (lines 174-175) |
|
|
|
|
|
Q18 |
: |
In line 145, please add information about the mixer used. |
|
R18 |
: |
I appreciate your suggestion. I have the added some information about the mixer in the manuscript (lines 177-178). |
|
|
|
|
|
Q19 |
: |
In line 148, please explain the selected hot pressing parameters, e.g. based on preliminary trials, other references, etc. |
|
R19 |
: |
I appreciate your reminder. I have explained some information to selected hot-pressing parameters in the manuscript (lines 180-188). |
|
|
|
|
|
Q20 |
: |
In line 166, please add the respective standard (SNI 03-2105-2006) in the manuscript references. The same comment applies to line 171 (ASTM D70), and line 181 (ASTM C177-9). In addition, I believe the correct standard in line 181 should be “ASTM C177 – 19. Standard Test Method for Steady-State Heat Flux Measurements and Thermal Transmission Properties by Means of the Guarded-Hot-Plate Apparatus”, please revise. |
|
R20 |
: |
I appreciate your reminding. I have revised the sentence in manuscript |
|
|
|
|
|
Q21 |
: |
In general, the Results and Discussion section is well-structured, concise, informative and discussed with previously published research works |
|
R21 |
: |
We would like to thank the reviewers for this statement |
|
|
|
|
|
Q22 |
: |
The Conclusions (lines 353-373) are consistent with the results and reflect the main findings of the study. |
|
R22 |
: |
We would like to thank the reviewers for this statement |
|
|
|
|
|
Q23 |
: |
The references cited are appropriate and correspond to the topic of the manuscript. The inclusion of additional references in all sections of the manuscript is highly recommended. This will significantly increase the value of the presented work |
|
R23 |
: |
We would like to thank the reviewers for this statement. We have added some relevant references according to the suggestions of reviewers. |

Round 2
Reviewer 1 Report
In my opinion, the manuscript is significantly improved.
The addition of more references in the introduction puts the research in the context of contemporary trends for obtaining eco-friendly materials.
The esteemed authors have complied with the recommendations made by mе. The choice of technological parameters and the obtained results are significantly clarified.